# Analysis of the Turbulent Lubrication of a Textured Hydrodynamic Journal Bearing

**Yazhou Mao [1,\*], Lilin Li [1], Daqing Li [1] and Jingyang Zheng [2]**

[1] School of Mechanical Engineering, Henan University of Engineering, Zhengzhou 451191, China
[2] College of Vehicle and Traffic Engineering, Henan University of Science and Technology, Luoyang 471003, China
\* Correspondence: myzhgy@haue.edu.cn

**Abstract:** In order to investigate the turbulent lubrication performance of a textured hydrodynamic journal bearing (THJB), a model of turbulent lubrication was established in this paper. The variations in the Reynolds number, oil film thickness, oil film pressure, bearing capacity, attitude angle, and side leakage flow with structural and working parameters were studied, and the axis whirl orbit was further analyzed. The results show that turbulent lubrication is suitable for the actual operating conditions of THJBs. The Reynolds number decreases with the eccentricity ratio in the pressure-bearing zone but increases with rotational speeds, whereas the variation in the maximum oil film pressure increases and the minimum oil film thickness decreases with the eccentricity ratio under various Reynolds numbers. The bearing capacity decreases with the dimple diameter, depth, oil film thickness, and clearance ratio but increases with the length/diameter ratio and dimple spacing. As the eccentricity ratio increases, the attitude angle decreases, but the side leakage flow increases. In addition, the system tends to be unstable as the rotational speed and length/diameter ratio increase, and the friction and wear on the surface are three-body friction. This work not only helps in analyzing the characteristics of a THJB under actual operating conditions but also provides support for research on the simulation of THJB's lubrication mechanism of THJB via computational fluid dynamics.

**Keywords:** lubrication performance; surface texture; turbulent regime; Reynolds number





## 1. Introduction

With the development of highly precise rotational machinery that can achieve high speed and tolerate heavy loads [1–3], as a key part that supports the rotating spindle, the lubrication performance of a hydrodynamic journal bearing (HJB) directly affects the system of rotating machinery. The surface texture can effectively make the lubrication performance of HJBs better [2,4,5], and the lubrication performance of textured bearings will be affected by the flow regime; laminar and turbulence are the common flow regimes that affect the properties of lubrication.

Until now, the development of laminar lubrication theory has been quite comprehensive. Turbulence is often encountered in natural science and engineering technology, but laminar lubrication, which has been studied well, is seldom encountered. In 1923, the viscous flow between concentric cylinders was studied, an expression for calculating the critical Reynolds number between concentric cylinders was proposed [6], and the application of turbulent lubrication theory in the design of friction pairs was deeply studied [7,8]. Since 1923, the lubrication theory has made a breakthrough. In particular, a turbulent Reynolds equation was established based on the Ng-Pan theory [9], and the lubrication performance parameters under a greater range of Reynolds numbers were obtained. The turbulent model was established to study the performance of turbulent lubrication, and the results are in good agreement with the corresponding experimental data [10–12]. While the finite difference method was used to solve a modified Reynolds equation of couple stress

fluids considering turbulence [13–15], the dynamic properties of water-lubricated bearings were analyzed under contact conditions, and it was found that the dynamic characteristics were significantly affected by speed and pressure [16]. Jiang's team conducted a series of studies, and the results showed that the influence of factors such as film cavitation, inertial force, and turbulence on the bearing lubrication performance could not be ignored under high-speed working conditions [17–19] and that surface topography has an impact on bearing performance [20,21]. Moreover, the lubrication performances of bearings with groove angles of 36° and 18° in a turbulent regime were analyzed, and the load capacity improved as the misalignment angle increased [22]. Further studies have shown that local turbulence has a significant impact on the friction coefficient [23,24], and the influence of micro-groove on lubrication characteristics and bearing capacity varies with variations of the eccentricity ratio [25,26]. The effect of the micro-groove on the lubrication performance of water-lubricated bearings during start-up was also studied [27–29].

A series of reports on lubrication properties have been analyzed by scholars, but there are few reports on the performance of micro pit THJBs under turbulent regimes. In this work, a systematic study of the influences of geometry and working condition parameters on the characteristics of micro pit THJBs was carried out by the authors. This research has a certain significance for our understanding of the performance laws and engineering applications of THJBs serving in turbulent regimes.

## 2. Geometry Model and Governing Equations

### 2.1. Geometry Model

As schematic and section of a THJB are shown in Figure 1. In Figure 1, $O_1$ and $O_2$ represent the center of THJB and the rotating shaft, respectively, $e$ is the eccentric distance, $R$ and $r$ represent the bearing radius and spindle radius, respectively, $c$ is the clearance $c = R - r$, $L$ is the bearing width, and $h_{\min}$ and $h_{\max}$ are the minimum and maximum oil film thickness, respectively.

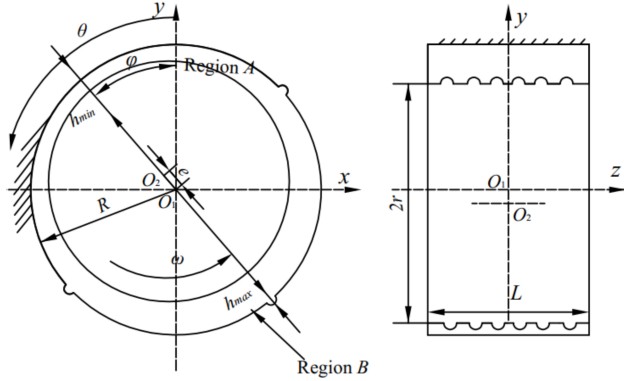

**Figure 1.** Schematic of textured hydrodynamic journal bearing.

The geometric parameters selected for the investigation of the lubrication characteristics of THJBs within a turbulent model are shown in Table 1.

**Table 1.** Geometric parameters of a textured hydrodynamic journal bearing.

| Parameters | Values |
| --- | --- |
| Bearing radius $R$/mm | 100 |
| Radial clearance, $c$/mm | 0.05–0.20 |
| Length/diameter ratio, $L/D$ | 1 |
| Eccentricity ratio, $\varepsilon$ | 0.25–0.95 |

**Table 1.** *Cont.*

| Parameters | Values |
|---|---|
| Lubricant viscosity, $\mu/\text{Pa·s}^{-1}$ | 0.0048 |
| Density, $\rho/\text{kg·m}^{-3}$ | 890 |
| Mass, $m/\text{kg}$ | 13.43 |
| Rotational speed, $n/\text{rpm}$ | 1000–20,000 |
| Texture radius, $r_\text{p}/\mu\text{m}$ | 16 |
| Texture depth, $h_\text{p}/\mu\text{m}$ | 25 |
| Texture density, $s_\text{p}/\%$ | 25 |

*2.2. Flow Regime Criterion*

The flow regime of a lubricating oil regime in the clearance of a bearing is related to its Reynolds number. The Reynolds numbers $Re_A$ and $Re_B$ of the THJB in region A and region B can be calculated using the following expression [6]:

$$Re_A = \rho \times u \times h / \mu \tag{1}$$

$$Re_B = \rho \times u \times h \times (1 - a \times \varepsilon) / \mu \tag{2}$$

where $\rho$ is the density, $u$ is the shaft velocity, $h$ is the oil film thickness, $\mu$ is the viscosity, and $a$ is the coefficient relating to the bearing's structural and working conditions, which can be obtained according to experimental data.

Expressions (1) and (2) show that the relationship between the Reynolds number, velocity, and oil film thickness, and the expression of the critical Reynolds number under turbulence can be obtained via estimation as follows [30]:

$$Re_\text{cc} = 41.1\sqrt{(1 + 1.5\varepsilon^2)/\psi} \tag{3}$$

where $\varepsilon$ is the eccentricity ratio, $\varepsilon = e/c$, $\psi$ is the clearance ratio, and $\psi = c/R$. The critical Reynolds number $Re_\text{cc}$ is obtained according to the expression in (3).

*2.3. Reynolds Governing Equation*

According to the flow regime analysis of the lubricant mentioned above, the Reynolds governing equation is as follows:

$$\frac{\partial}{\partial x}\left(\frac{h^3 G_\theta}{\mu}\frac{\partial p}{\partial x}\right) + \frac{\partial}{\partial z}\left(\frac{h^3 G_\lambda}{\mu}\frac{\partial p}{\partial z}\right) = 6R\omega\frac{\partial h}{\partial x} + 12\frac{\partial h}{\partial t} \tag{4}$$

where $x$ and $z$ represent two coordinates of a cylindrical coordinate system, $\omega$ represents the shaft angular velocity, $p$ represents the oil film pressure, and $G_\theta$ and $G_\lambda$ represent turbulence factors.

Considering the convenience of the analysis process and the simplicity of the governing equation, the dimensionless, unified Reynolds governing equation, which is suitable for analyzing the turbulent lubrication of THJBs, is provided below:

$$\frac{\partial}{\partial \theta}\left(H^3 G_\theta \frac{\partial P}{\partial \theta}\right) + \left(\frac{D}{L}\right)^2 \frac{\partial}{\partial \lambda}\left(H^3 G_\lambda \frac{\partial P}{\partial \lambda}\right) = \frac{\partial H}{\partial \theta} + 2\dot{e}\cos\theta \tag{5}$$

where $H$ and $P$ represent the normalized oil film thickness and pressure, respectively, $D$ and $L$ represent the diameter and length of the bearing, respectively, and $\lambda$ and $\theta$ represent the normalized circumferential coordinates and axial coordinates, respectively.

In the dimensionless process of the Reynolds equation [4], the following dimensionless parameters are introduced:

$$\left.\begin{array}{l} x = R\theta \\ h = Hc \\ z = \lambda L/2 \\ p = Pp_0 \\ p_0 = 6\mu\omega/\psi^2 \end{array}\right\} \tag{6}$$

Based on the Ng-Pan mathematical model of lubrication [10], the turbulence factors $G_\theta$ and $G_\lambda$ in Equation (4) are $G_\theta = 1/(12 + 0.0136Re_{cc}^{0.9})$ and $G_\lambda = 1/(12 + 0.0043Re_{cc}^{0.98})$, respectively, while the oil film thickness of the THJB mainly consists of untextured and textured oil film thicknesses, and its expression for the THJB is shown as follows:

$$H = 1 + \varepsilon\cos(\theta - \varphi) + \sum_{i=1}^{n} H_i \tag{7}$$

where $H_i$ represents the oil film thickness of the micro-pits and $i$ represents the number of micro-dimples.

The Reynolds boundary condition of the Reynolds governing equation was adopted. In this work, the oil film pressure of the first column and column ($m + 1$) were set to zero as a starting boundary. The process of realizing the oil film's pressure rupture boundary was as follows: The pressure along the circumferential direction is calculated point by point from $j = 2 \sim n$ rows and $i = 2 \sim m$ columns (see Figure 2). If the pressure value is negative, it is determined that oil film rupture occurs at this point, and the pressure values at this point and the point after its row are set to zero. After an iterative calculation, the rupture boundary gradually approaches the natural rupture boundary, and the pressure distribution meets the Reynolds boundary condition.

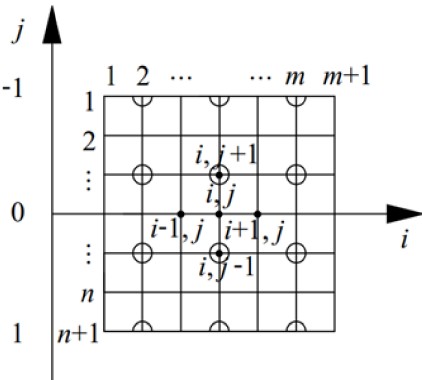

**Figure 2.** Grid division of a textured hydrodynamic journal bearing.

*2.4. Bearing Capacity and Attitude Angle Equation*

As the oil film force comes into balance with the normal force, the oil film component forces of the THJB are $W_x$ and $W_y$, respectively. Therefore, the expression of the bearing's capacity is as follows:

$$W_x = \int_0^{2\pi} \int_0^{L} P\cos\theta\lambda d\theta d\lambda$$

$$W_y = \int_0^{2\pi} \int_0^{L} P\sin\theta\lambda d\theta d\lambda \tag{8}$$

$$W = \sqrt{W_x^2 + W_y^2}$$



Thus, the attitude angle of the THJB is shown as follows:

$$\varphi = \text{arctg}\frac{W_y}{W_x} \tag{9}$$

### 2.5. Lubricant Side Leakage Flow Equation

For THJBs, the velocity components in the circumferential direction and the axial direction are as follows:

$$V_x = \frac{1}{2\mu}\frac{\partial p}{G_\theta \partial x}\left(y^2 - hy\right) + \frac{U}{h}y$$

$$V_z = \frac{1}{2\mu}\frac{\partial p}{G_\lambda \partial z}\left(y^2 - hy\right) \tag{10}$$

According to the flow equation, Equation (10) is integrated, and the flow on any given section is obtained as follows:

$$q_x = -\frac{h^3\partial p}{12\mu G_\theta \partial x} + \frac{Uh}{2}$$

$$q_z = -\frac{h^3\partial p}{12\mu G_\lambda \partial z} \tag{11}$$

Thus, the dimensionless expression of the obtained lubricant side leakage flow is shown as follows:

$$Q = \frac{-RL\int_0^{2\pi}\frac{H^3c^3}{\mu}\frac{p_0\partial P}{G_\lambda\partial(\lambda L)}\Big|_{\lambda=1}d\varphi}{6r^2cU} = -\int_0^{2\pi}H^3\int_0^{2\pi}\frac{\partial P}{G_\lambda\partial\lambda}\Big|_{\lambda=1}d\varphi \tag{12}$$

### 2.6. Axis Whirl Orbit Governing Equation

Axis whirl orbit analysis is an important way to determine the stability of the rotor. The governing equation of the axis whirl orbit for a rotor–THJB system is shown as follows.

$$\begin{cases} m\ddot{x} = W_x + me\omega n^2\sin(\omega t) \\ m\ddot{y} = W_y + me\omega n^2\cos(\omega t) + mg \end{cases} \tag{13}$$

where $e$ is the centrifugal distance, $\omega$ is the speed, $W_x$ and $W_y$ are the oil film forces in $x$-axis and $y$-axis directions, respectively, $g$ is the acceleration of gravity, and $x$ and $y$ are the displacements of journal in $x$ axis and $y$ axis directions, respectively.

The expressions $W_x$ and $W_y$ of the oil film force components on the journal could be obtained by solving the Reynolds governing equation, and the instantaneous acceleration of the journal was obtained via the following expression.

$$\ddot{x} = W_x/m$$

$$\ddot{y} = \left(W_y - W\right)/m \tag{14}$$

Then, the expressions for the transient velocity and displacement of the axis were

$$\dot{x}_i = \ddot{x}_i\Delta t + \dot{x}_{i-1}$$

$$\dot{y}_i = \ddot{y}_i\Delta t + \dot{y}_{i-1}$$

$$x_i = \dot{x}_i\Delta t + x_{i-1}$$

$$y_i = \dot{y}_i\Delta t + y_{i-1} \tag{15}$$

where $\Delta t$ is a tiny time interval, $\Delta t = 120/50n$, and $n$ is the rotational speed. The axis whirl orbit of the journal at any time was obtained via Expressions (13)–(15).

### 3. Numerical Computation Method and Effectiveness Analysis

In this work, the effectiveness was verified prior to conducting a numerical analysis. In order to verify the correctness of the finite difference solution, the turbulent lubrication performance of a THJB with the geometric parameters $\psi = 0.003$ and $L/D = 1$ was analyzed, and the Sommerfeld number, a characteristic parameter, was obtained via calculation and compared with results in the literature [31]. The Sommerfeld number is:

$$S = \frac{\mu L u}{\pi W} \left( \frac{R}{c} \right)^2 \tag{16}$$

The results of comparing the solution obtained via FDM with test values are shown in Figure 3. In Figure 3, the calculations agree well with test values, which proves the correctness of the FDM applied in this work.

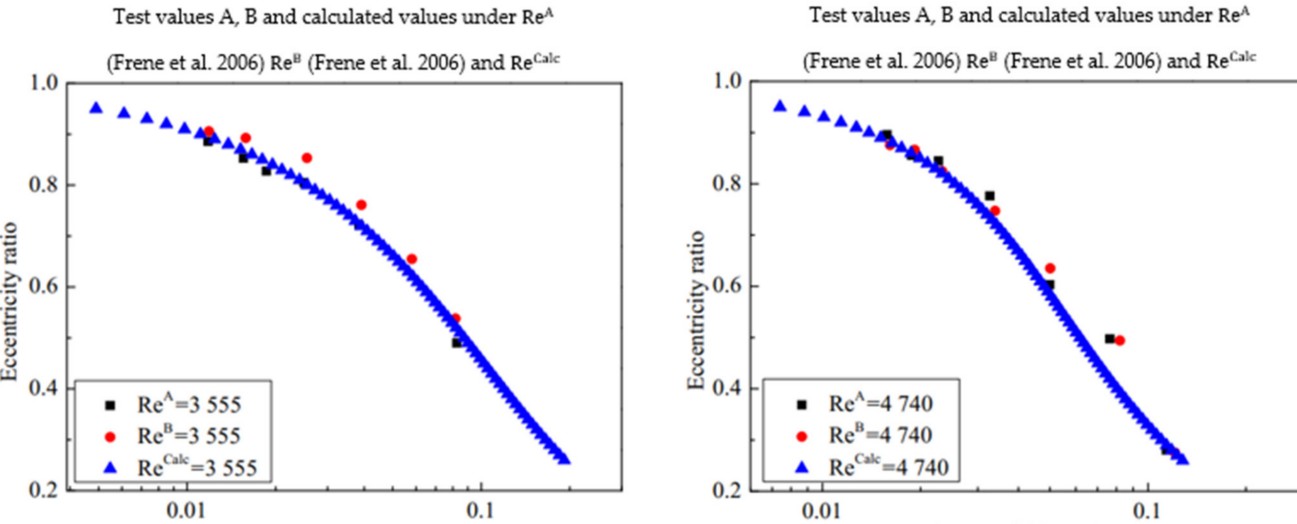

**Figure 3.** The variation of the eccentricity ratio with Sommerfeld number [31].

Equation (5) was solved by FDM, and the grid number was important when ensuring the correctness of the calculation result. In order to identify a suitable differential grid, the properties of a THJB were analyzed using various differential grids based on the geometric parameters in Table 1. The results are listed in Table 2. Relative errors of oil film pressure are 0.54% and 1.48%, respectively, for the 390 × 125 differential grid compared with the 628 × 200 differential grid, while the relative errors are 0.03% and 0.15%, respectively, in [32,33] compared with the FDM result achieved using the same differential grid, 314 × 100. Therefore, the 314 × 100 differential grid is used in this paper.

**Table 2.** Comparison results for various differential grids.

| Differential Grid | Maximum Oil Film Pressure $P_{max}$/MPa | | | | |
|---|---|---|---|---|---|
| | Sun et al. [32] | Lv et al. [33] | FDM | Errors 1 | Errors 2 |
| 314 × 100 | 33.06 | 33.0 | 33.05 | 0.03% | 0.15% |
| 390 × 125 | 33.06 | 33.0 | 32.87 | 0.58% | 0.39% |
| 628 × 200 | 33.06 | 33.0 | 32.56 | 1.54% | 1.35% |

In order to further verify the accuracy of calculation results based on the turbulent lubrication model, a comparison of the theoretical result, test value result, and the results from other studies in the literature [34,35] for the oil film pressure distribution are shown in Figure 4. The results show that the difference between the theoretical result presented

in this paper, the test value, and the result from other studies are relatively small, which proves the accuracy of the calculation result based on the turbulent model.

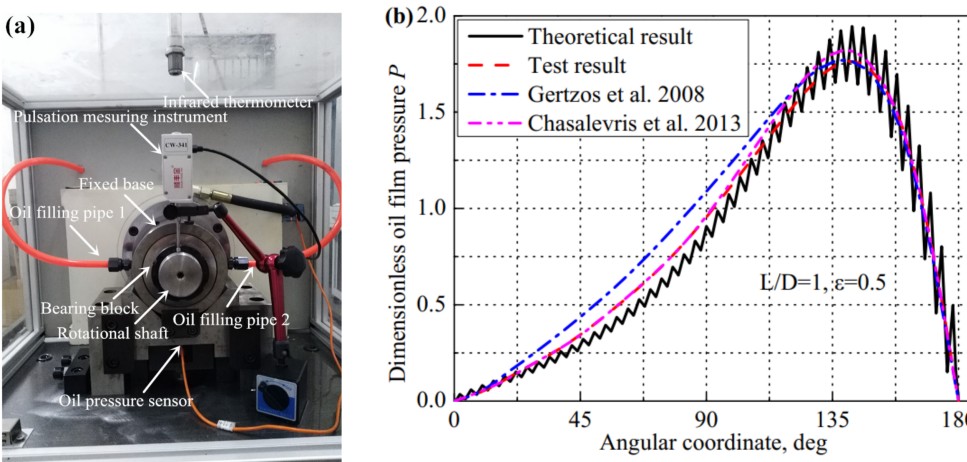

**Figure 4.** Comparison of oil film pressure distribution theoretical result, test value, and results from other studies in the literature. (**a**) Oil film pressure test bench; (**b**) Comparison of oil film pressure distribution [34,35].

## 4. Results and Discussion

### 4.1. Lubrication Characteristics of a Textured Bearing under the Turbulent Model

#### 4.1.1. Reynolds Number Varies with Eccentricity Ratios and Rotational Speeds

The variation in the Reynolds number with eccentricity ratios ($\varepsilon$ = 0.25, 0.50, 0.75, and 0.95, $n$ = 1000 rpm) and rotational speeds ($\varepsilon$ = 0.25, $n$ = 200, 400, 600, 800, 1000, 1200, and 1400 rpm) is shown in Figure 5. In Figure 5a, the Reynolds number decreases with an increase in the eccentricity ratios in the pressure-bearing region. In contrast, the Reynolds number increases with eccentricity ratios, and the Reynolds number for the pressure-bearing region is higher than the critical Reynolds number; however, the opposite is true for non-pressure-bearing regions. This is because the oil film thickness in the pressure-bearing zone is relatively thinner, and the mainstream region where the Reynolds number is lower than the critical Reynolds number is laminar. Conversely, the mainstream region of the lubricating oil is turbulent where the flow states are more complex. In Figure 5b, the Reynolds number increases with the increase in the rotational speed $n$, and the actual Reynolds number is less than the critical Reynolds number when $n$ = 200–800 rpm. When $n$ = 1000–1200 rpm, the actual Reynolds number is partially less than the critical Reynolds number, and the others are higher than the critical Reynolds number. However, when $n$ is 1400 rpm, the Reynolds number is higher than the critical Reynolds number. The reason for this is that the flow regimes of lubricating oil are laminar and the mixture flows when $n$ = 200–800 rpm and 1000–1200 rpm, respectively. In addition, the flow regime of lubricating oil is turbulent when $n$ is 1400 rpm.

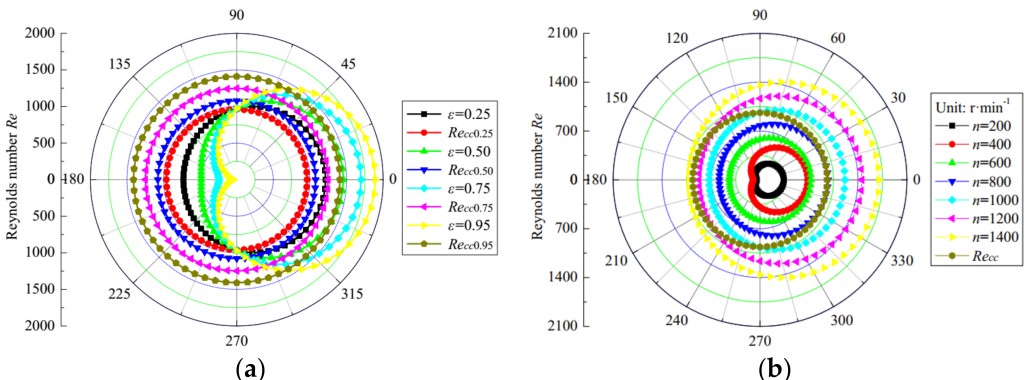

**Figure 5.** Reynolds number varies with eccentricity ratios and rotational speeds. (**a**) Reynolds number varies with eccentricities; (**b**) Reynolds number varies with rotational speeds.

### 4.1.2. Variations in Oil Film Thickness and Maximum Oil Film Pressure

Variations in oil film thickness and pressure with eccentricity ratio and angular coordinate are shown in Figure 6. In Figure 6a, the trend in the variation in the oil film thickness is basically consistent and shows a sine wave shape under six eccentricity ratios. As the eccentricity ratio increases, the minimum oil film thickness decreases gradually. The reason for this is that as the eccentricity ratio increases, the load on the rotor increases and the offset of the axial center position increases, which causes the oil film thickness to decrease. In Figure 6b, the minimum oil film thickness decreases with the increase in the eccentricity ratio and follows the rule of $H = -\varepsilon$, while the maximum oil film pressure increases gradually with eccentricity ratios. In addition, the minimum oil film thickness is not affected by the variation in the Reynolds numbers under the same eccentricity, and the maximum oil film pressure is almost unaffected by the eccentricity ratios (<0.25) and Reynolds numbers. The larger the Reynolds number at the same eccentricity ratio, the greater the corresponding maximum oil film pressure.

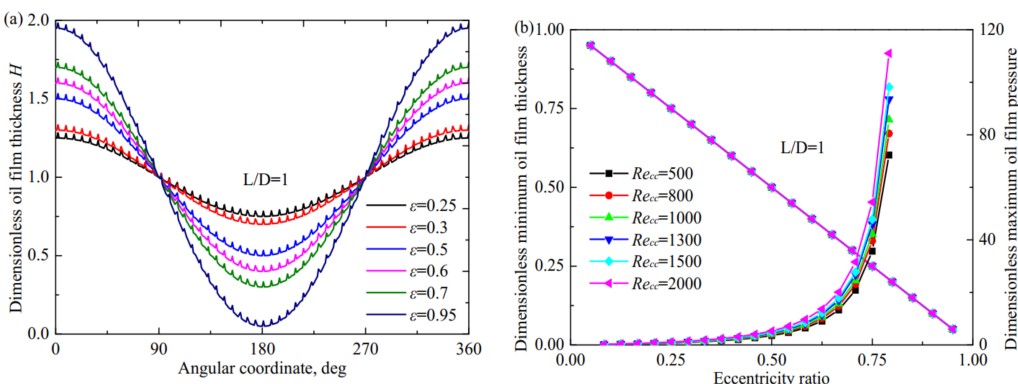

**Figure 6.** Variations in the oil film thickness and pressure with eccentricity and angular coordinate. (**a**) The variation of oil film thickness; (**b**) The variation of minimum oil film thickness and maximum oil film pressure with eccentricity ratio.

### 4.1.3. Variation in Oil Film Pressure Distribution

The oil film distribution of a THJB with a length-to-diameter ratio $L/D = 1$ and an eccentricity ratio $\varepsilon = 0.3$ is shown in Figure 7. In Figure 7a, the circumferential oil film pressure distribution is basically the same, but the oil film pressure under a turbulent regime is much higher than under the laminar regime, which indicates that a turbulent regime can effectively make the oil film pressure better. In order to observe the pressure distribution of the THJB under a turbulent regime, the variation in the oil film pressure is shown in Figure 7b. In Figure 7b, the smallest and largest THJB oil film pressures are

located in the leaking position and the axial center position $X = 0$, respectively. Moreover, the oil film pressure is symmetrically distributed along the axial direction, with $X = 0$ as the axis of symmetry.

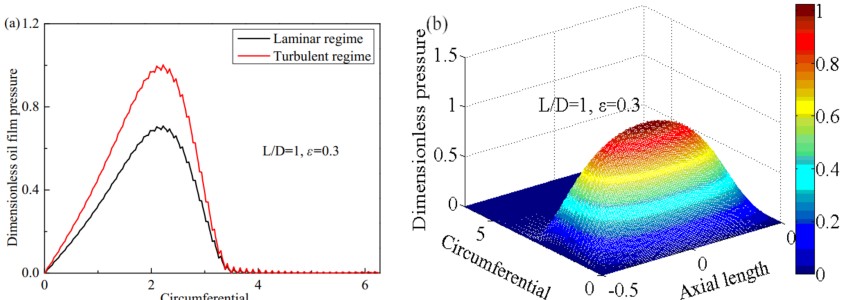

**Figure 7.** The variation in oil film pressure distribution with $L/D = 1$ and $\varepsilon = 0.3$. (**a**) The variation of oil film pressure under various flow regimes; (**b**) The distribution of oil film pressure.

The circumferential and axial oil film pressure distributions of the THJB are shown in Figure 8. In Figure 8a, the variation in the oil film pressure distribution becomes serrated due to the existence of surface texture, and arbitrary serrations correspond to the texture point. The dynamic lubrication of the THJB system is superimposed, which further enhances the oil film pressure as the surface texture is in the convergence region. Therefore, the serrated change is not obvious. In contrast, in the divergent region, the negative pressure effect of the surface texture will truncate the dynamic pressure action in the convergence region, which will cause the cavitation phenomenon to be more obvious and the dynamic pressure lubrication effect caused by the gap contraction to be more prominent so that the serration presents a significant change. In Figure 8b, the axial oil film pressure distribution shows a continuous phenomenon of alternating changes between high-pressure and low-pressure regions. The maximum and minimum oil film pressure values occur at the axial center coordinate ($X = 0$) and the axial leaking position ($X = \pm 0.5$), respectively.

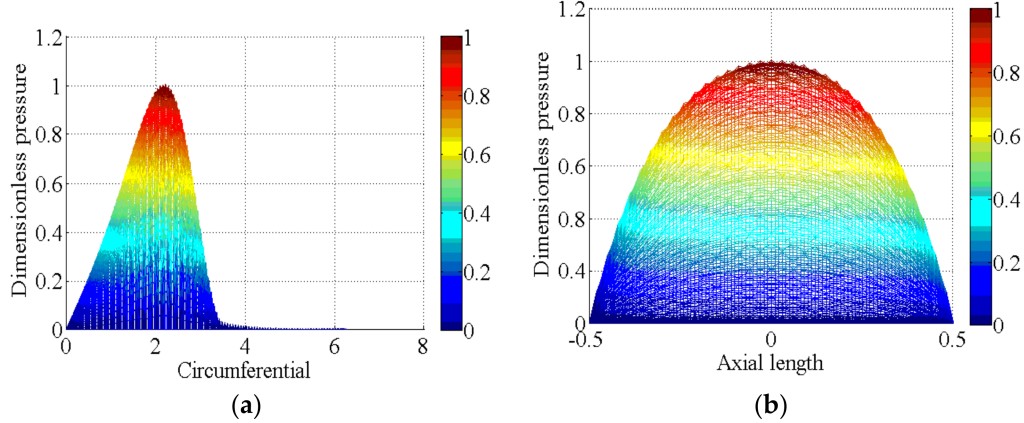

**Figure 8.** Sectional view oil film pressure of a textured bearing; (**a**) Circumferential oil film pressure; (**b**) axial oil film pressure.

In order to further explain the variation in the pressure distribution of THJB, the local oil film pressure distribution is shown in Figure 9. In Figure 9a, there is a zone in which the relative oil film pressure increases around the drop zone of the THJB under a turbulent regime. This is because the distribution interval of surface texture will lead to the local film thickness increasing as shown in Figure 9b. According to the hydrodynamic lubrication mechanism, an increase in the oil film thickness in the local area will inevitably lead to a decrease in the oil film pressure; thus, the situation in which the oil film pressure is

reduced is shown in Figure 9a. However, as the lubricating oil in the micro-pit texture flows out, the clearance becomes smaller and overlaps with the hydrodynamic effects of the THJB. Compared with the area in which the inner oil film pressure of the micro pit decreases, the outside has a higher oil film pressure increase area, that is, the low-pressure area appears in front of the surface texture the and high-pressure area appears in the rear.

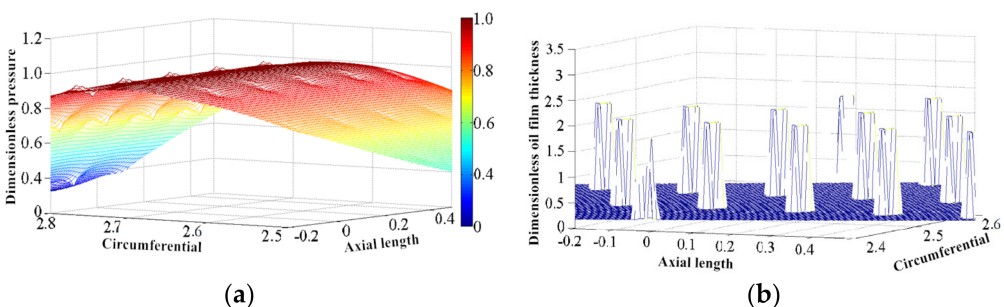

(**a**)                               (**b**)

**Figure 9.** Local oil film pressure distribution of textured bearing: (**a**) Local oil film pressure distribution; (**b**) local oil film thickness distribution.

### 4.2. The Influence of Structural and Working Condition Parameters

The bearing capacity distribution of a THJB is influenced by the micro dimple diameter, depth, spacing, and oil film thickness, as shown in Figure 10. In Figure 10a,b, the bearing capacity increases gradually as the Reynolds number increases with the same dimple diameter and depth. Moreover, a higher Reynolds number is beneficial to improve the bearing capacity. Therefore, the Reynolds number is a very important factor affecting the bearing capacity.

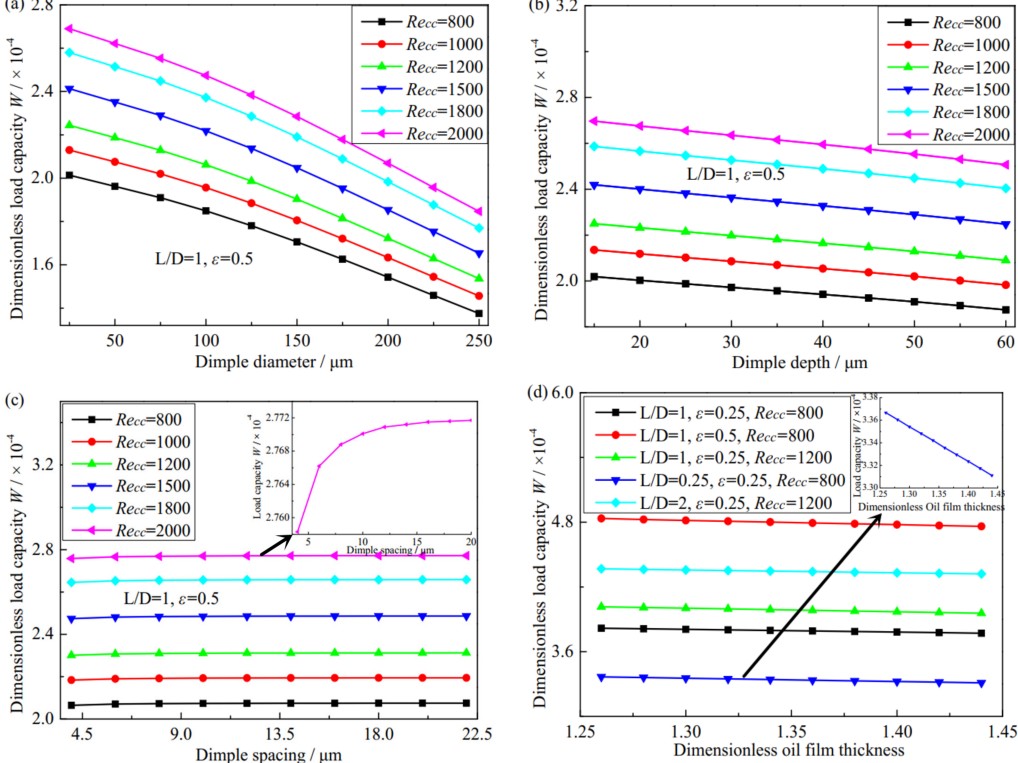

**Figure 10.** Variation in bearing capacity with dimple diameter, depth, spacing, and film thickness. (**a**) The variation of load capacity with dimple diameter; (**b**) The variation of load capacity with dimple depth; (**c**) The variation of load capacity with dimple spacing; (**d**) The variation of load capacity with oil film thickness.

In Figure 10c, the bearing capacity of the THJB increases gradually as the dimple spacing increases. The reason for this is that the adjacent micro pits do not interfere with each other when the spacing is small. However, the adjacent micro pits experience a mutual disturbance with an increase in the spacing, which causes the change rule to increase slowly. Additionally, the bearing capacity increases gradually as the Reynolds number increases with the same dimple spacing. Thus, a larger load capacity is obtained by changing the dimple spacing and Reynolds number.

Moreover, in Figure 10d, there is an approximately negative linear relationship between the oil film thickness and bearing capacity. The bearing capacity is approximately twice that of 1.44 as the oil film thickness is 1.26. Additionally, the smaller the eccentricity ratio, Reynolds number, and length/diameter ratio, the smaller the bearing capacity. However, the maximum bearing capacity of the THJB occurs at a length-to-diameter ratio of $L/D = 1$, an eccentricity ratio $\varepsilon = 0.5$, and a Reynolds number of $Re_{cc} = 800$. Therefore, the load capacity will be affected by the geometric parameter and Reynolds number.

The variation in the bearing capacity of the THJB with the length/diameter ratio and clearance ratio is shown in Figure 11. In Figure 11a, the bearing capacity increases as the length/diameter ratio increases. If the length/diameter ratio is too large, its influence on the bearing capacity is almost unchanged. The reason for this is that with the increase in the length/diameter ratio, although the axial width increases and the bearing range increases, the pressure per unit area decreases, so the bearing capacity decreases as the length-to-diameter ratio increases. Therefore, the length/diameter ratios of THJBs should be selected in the range of 1–2.4. The bearing capacity increases with an increase in the eccentricity ratio under the same Reynolds number and the Reynolds number under the same eccentricity ratio. In addition, the bearing capacity of a THJB with a Reynolds number of $Re_{cc0.3} = 1500$ is better than that of a textured bearing with Reynolds numbers of $Re_{cc0.25} = 1000$ and $Re_{cc0.25} = 1200$. However, in Figure 11b, the bearing capacity decreases with an increase in the clearance ratio but increases gradually as the Reynolds number increases under the same clearance ratio, and a higher Reynolds number is beneficial to improve the bearing capacity.

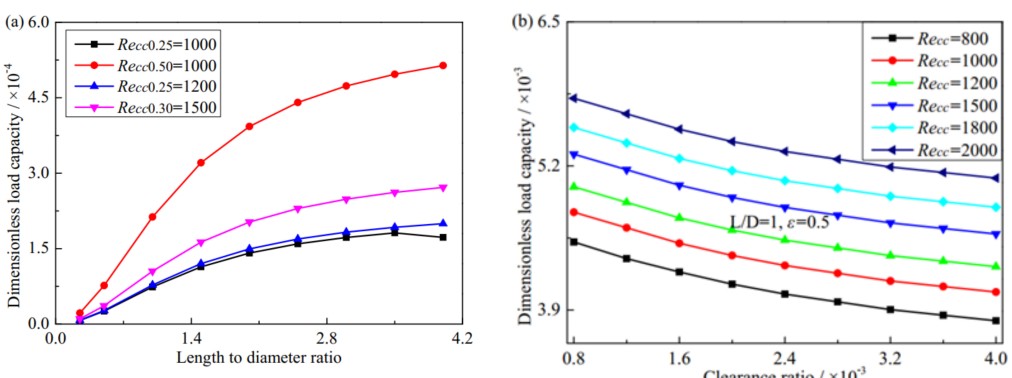

**Figure 11.** Variation in bearing capacity with length/diameter ratio and clearance ratio. (**a**) Bearing capacity with length to diameter ratio; (**b**) Bearing capacity with clearance ratio.

The distribution of the bearing capacity of a THJB with rotational speed is shown in Figure 12. In Figure 12a, the linear increase in the bearing capacity with an increasing rotational speed occurs because the increase in rotational speed changes the THJB's oil film pressure distribution. However, at lower rotational speeds and at the same rotational speed, the bearing capacity of the THJB under a turbulent regime is approximately the same as that of a THJB under a laminar regime because, at this speed, the flow regime gradually changes from laminar to turbulent. In Figure 12b, the bearing capacity becomes greater as the length/diameter ratio increases with the increase in the rotational speed under the same Reynolds number and eccentricity ratio. However, under the same length/diameter ratio and eccentricity ratio, the bearing capacity increases as the Reynolds number increases

with the increase in the rotational speed, for example, $Re_{cc}$ = 600–1200. In addition, the greater eccentricity, the greater the bearing capacity as the rotational speed increases under the same Reynolds number and length/diameter ratio.

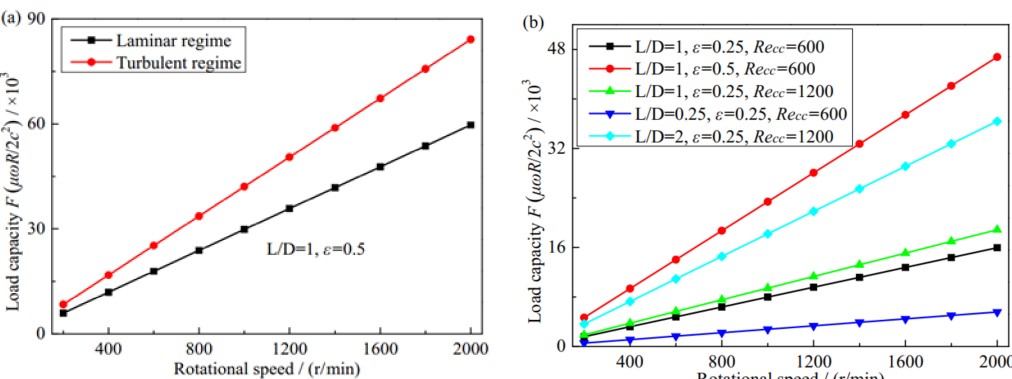

**Figure 12.** Variation in bearing capacity with rotational speed. (**a**) Bearing capacity with rotational peeds under various flow regimes; (**b**) Bearing capacity with rotational speeds under various parameters.

### 4.3. Variations in the Attitude Angle and Lubricant Side Leakage Flow with the Eccentricity Ratio

The variations in the attitude angle and lubricant side leakage flow with the eccentricity ratio are shown in Figure 13. In Figure 13a, the attitude angle decreases as the eccentricity ratio increases, which means that bearing capacity increases as the eccentricity ratio increases, and the oil film's bearing capacity increases as the attitude angle decreases. In addition, when extreme conditions occur, since the eccentricity ratio $\varepsilon$ tends toward 1, the attitude angle $\varphi$ tends toward 0°. This is because as eccentricity $\varepsilon$ = 1, the shaft and bearing bush are in direct contact, the direction of the bearing capacity is along the connecting line between the shaft diameter and bearing bush, and the corresponding attitude angle is 0°.

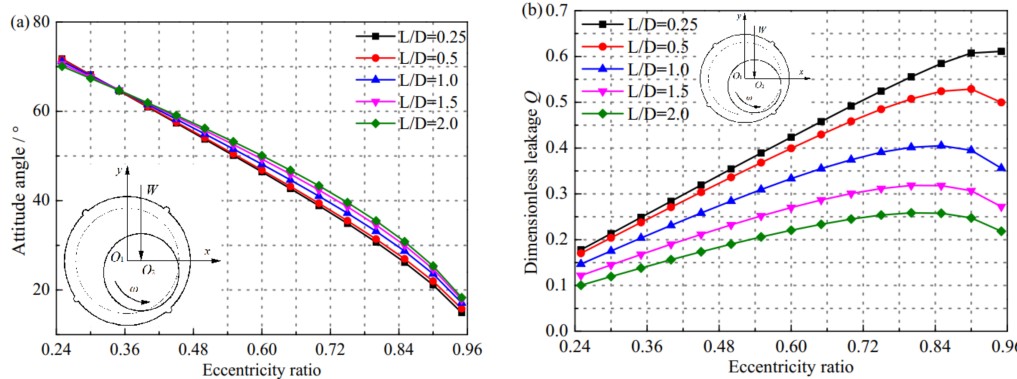

**Figure 13.** Variation in the attitude angle and leakage with eccentricity ratio. (**a**) Attitude angle with eccentricity ratio; (**b**) Leakage with eccentricity ratio.

In Figure 13b, the dimensionless lubricant's side leakage flow increases with the increase in the eccentricity ratio. This occurs because the lubricating oil's side leakage flow is determined by the oil film pressure gradient in the axial direction of the THJB. As the eccentricity ratio increases, the oil film's pressure gradient also increases at both ends of the THJB. Therefore, the dimensionless lubricating oil's side leakage flow increases as the eccentricity ratio increases, whereas when the eccentricity ratio tends toward 1 due to dry friction occurring between the shaft diameter and bearing bush, and when good hydrodynamic lubrication cannot be established, no oil film pressure gradient will be generated. Thus, the dimensionless lubricating oil's side leakage flow tends toward zero at this time. Additionally, as the length/diameter ratio increases, the THJB's lubricating

oil's side leakage flow decreases. This is because the axial flow resistance increases as the length/diameter ratio increases, which leads to decreases in the pressure gradient.

### 4.4. The Axis Whirl Orbit Variation of the Rotor

The axis whirl orbit variation of the rotor calculated based on Section 2.6 is shown in Figure 14. In Figure 14a,b, the rotor is initially located at the center of the THJB, and offset has not occurred due to the rotational spindle. The rotor begins to move downward due to the action of gravity, and the pressure generated by the oil film gradually increases as the eccentricity ratio increases. When the pressure generated by the oil film is higher than the rotor's gravity, the rotor starts to move upward, and the running area gradually converges and tends toward a stable state. In Figure 14a,d, with the increase in the length/diameter ratio, the axis whirl orbit of the rotor is a limit cycle, and the rotor is in a critical state at this time. Additionally, in Figure 14a,c, as the speed increases, the divergent state is occupied by the axis whirl orbit. Compared with the nonlinear axis whirl orbit when the rotational speed is 1000 rpm, the rotor axis whirl orbit diverges significantly outward. The reason for this is that the oil film flow regime tends to become more complex when the rotational speed increases, the oil film force on the rotor increases, and the attitude angle of the bearing capacity is larger, so the axis whirl orbit of the rotor is in a divergent state.

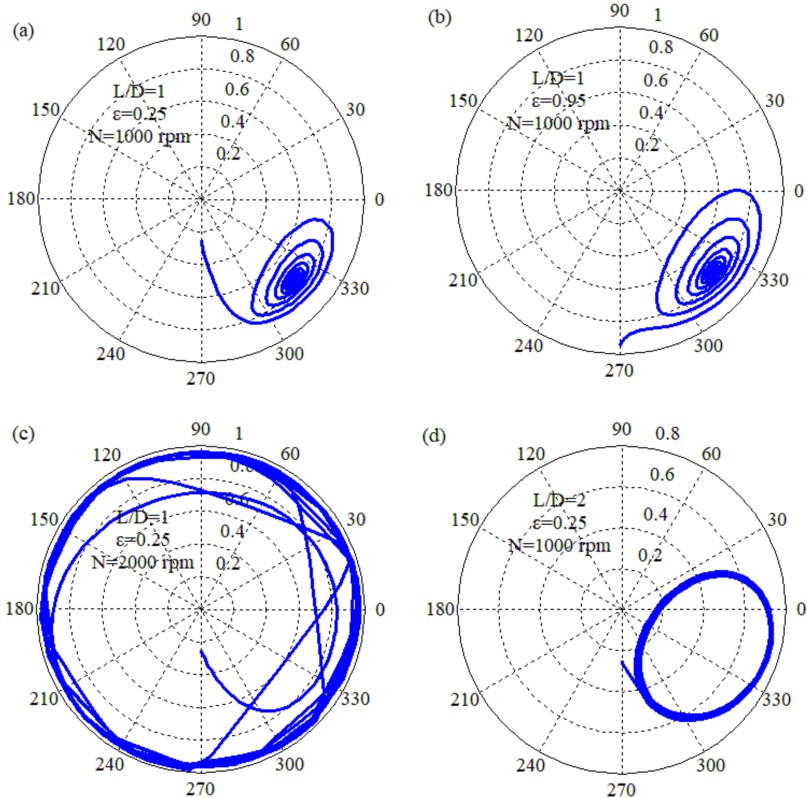

**Figure 14.** The variation in the axis whirl orbit of the rotor. (**a**) The axis whirl orbit variation of the rotor under $L/D = 1$, $\varepsilon = 0.25$ and N = 1000 rpm; (**b**) The axis whirl orbit variation of the rotor under $L/D = 1$, $\varepsilon = 0.95$ and N = 1000 rpm; (**c**) The axis whirl orbit variation of the rotor under $L/D = 1$, $\varepsilon = 0.25$ and N = 2000 rpm; (**d**) The axis whirl orbit variation of the rotor under $L/D = 2$, $\varepsilon = 0.25$ and N = 1000 rpm.

Figure 15 shows the friction and wear surface morphology of the THJB when the load and rotational speed were 4 kN, 1100 rpm, and 2000 rpm, respectively. In Figure 15, obvious scratches can be seen on the surface of the THJB. Apart from the micro pits being smoothed out, most of the micro pit textures were filled with wear particles. Because the rotor-bearing system was working at high speeds and under a heavy load, the lubricating

oil was not supplied in time, and the particles produced by friction were not removed effectively, which caused obvious grinding and filling of furrows.

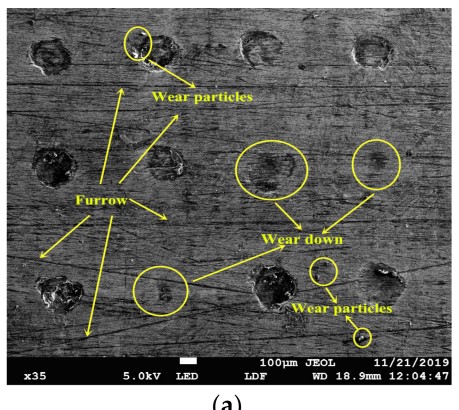 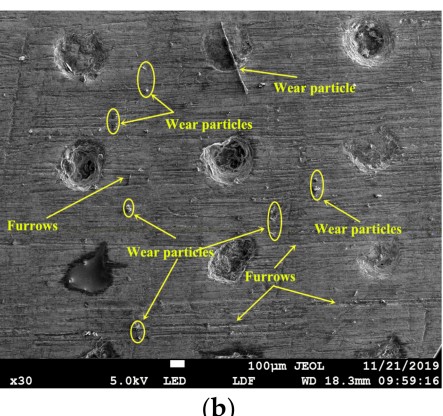

(**a**)                                    (**b**)

**Figure 15.** Friction and wear surface morphology of the THJB following operation as various rotational speeds; (**a**) SEM morphology of surface after operation at 1000 rpm; (**b**) SEM morphology of surface after operation at 2000 rpm.

## 5. Conclusions

In this present work, a model of the turbulent lubrication properties of a THJB was investigated. A few conclusions are summarized as follows:

(1)    A turbulent lubrication model was established for a performance analysis based on the Ng-Pan model, and the correctness of the model and the numerical method was verified. For the THJB in this work, the flow regimes were laminar and mixed flow at rotational speeds of $n$ = 200~800 rpm and 1000~1200 rpm, whereas a completely turbulent regime occurred at $n \geq 1400$ rpm. The difference between the values calculated for the laminar and turbulent model shows that the turbulent effect enhanced the dynamic pressure effect of lubricating oil, which cannot be ignored when designing textured bearings.

(2)    The Reynolds number decreases with the eccentricity in the pressure-bearing zone but increases with the rotational speed. The variation in the maximum oil film pressure increases and the minimum oil film thickness decreases with the eccentricity ratio under various Reynolds numbers. Furthermore, the bearing capacity of the THJB under a turbulent regime decreases with the dimple diameter, depth, oil film thickness, and clearance ratio but increases with the dimple spacing and length/diameter ratio. In addition, the attitude angle decreases but the side leakage flow increases as the eccentricity ratio increases.

(3)    The axis trajectory converged to an equilibrium point as the eccentricity values were $\varepsilon$ = 0.25 and 0.95 under the same rotational speed $n$ = 1000 rpm. Additionally, the axis trajectory spread outward and gradually became unstable as the rotational speed increased under the same eccentricity $\varepsilon$ = 0.25. Further study showed that the mechanism of friction and wear during the axis trajectory operation was mainly three-body friction. Moreover, the axis trajectory approached the cycle limit gradually as the length/diameter ratio changed from 1 to 2, and the length/diameter ratio should be properly selected to ensure the stable operation of the rotor when designing the THJB.

**Author Contributions:** Conceptualization, Y.M. and L.L.; methodology, Y.M., L.L. and D.L.; software, J.Z. and D.L.; formal analysis, L.L., D.L. and J.Z.; investigation, L.L. and J.Z.; resources, D.L. and J.Z.; data curation, Y.M., L.L. and D.L.; writing—original draft preparation, Y.M. and L.L.; writing—review and editing, Y.M. and D.L.; supervision, L.L.; project administration, Y.M., D.L. and J.Z.; funding acquisition, D.L. and J.Z. All authors have read and agreed to the published version of the manuscript.

**Funding:** This work was supported by the Henan Provincial Department of Science and Technology Research Project (No. 232102220041) and the Young Elite Scientists Sponsorship Program (No. 2023HYTP012).

**Data Availability Statement:** The data that support the findings of this study are available from the corresponding author.

**Acknowledgments:** The authors would also like to express their sincere for their constructive comments.

**Conflicts of Interest:** The authors declare no conflict of interest.

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
