# Peer review of "Analysis of the Turbulent Lubrication of a Textured Hydrodynamic Journal Bearing"

_lubricants, doi:10.3390/lubricants11090362_

Round 1

Reviewer 1 Report

The authors have established the turbulent model lubrication model to investigate lubrication performance of textured hydrodynamic journal bearing (THJB) under turbulent regime. The research content of the paper is relatively rich. However, the innovation points of the paper need to be highlighted again in the introduction section. A few technical aspects were unclear and below I have summarized my major points which I would expect the authors can adequately answer.  I include a few suggestions below.

(1) The conclusion section in the abstract should be carefully revised, as it feels unclear and difficult to understand. Please correct the expression “In order to investigated” as “In order to investigate”.

(2) What are the physical meanings of some variables in the text, such as Texture radius rp, ??A. What is the relationship with ??B and ?? in the paper?

(3) In Equation (3), the critical Reynolds number is only related to the eccentricity. Please confirm whether it is correct?

(4) How to measure the circumferential water film pressure in the experiment in Figure 3?

(5) Is the variation of the axis whirl orbit of rotor in Figure 13 obtained from experiments or calculations? There is no time domain calculation method for the axis whirl orbit of rotor mentioned in the paper.

It is important to notice that a major improvement of the English presentation of the work should take place.

Author Response

Dear Reviewer:

Thank you for your letter and for the your comments concerning our manuscript entitle” Analysis of the Turbulent Lubrication of a Textured Hydrodynamic Journal Bearing” (ID:lubricants-2532008). Those comments are all valuable and very helpful for revising and improving our paper, as well as the important guiding significance to our researches. We have studied comments carefully and have made correction which we hope meet with approval. The main corrections in the paper and the responds to the reviewer’s comments are as following:

Responds to the reviewer’s comments:

Reviewer #1:

  1. Response to comment: The conclusion section in the abstract should be carefully revised, as it feels unclear and difficult to understand. Please correct the expression “In order to investigated” as “In order to investigate”.

Response: Thank you for pointing this problem. The authors have revised the the expression “In order to investigated” as “In order to investigate”, as detailed in the conclusion section of the abstract and the authors have marked bold.

  1. Response to comment: What are the physical meanings of some variables in the text, such as Texture radius rp, A. What is the relationship with B and  in the paper?

Response: Thank you for pointing this question. As mentioned in this paper, the physical meanings of some variables are shown in Table 1. However, for the relationship between A and B, which represents a flow regime transition from one state to another.

  1. Response to comment: In Equation (3), the critical Reynolds number is only related to the eccentricity. Please confirm whether it is correct?

Response: Thank you for the reviewer’s kind advice. In Equation (3), the critical Reynolds number is not only related to the eccentricity ε, but also related to the clearance ratio ψ, where ψ = c/R is explained in detail under equation (3).

  1. Response to comment: How to measure the circumferential water film pressure in the experiment in Figure 3?

Response: Thank you for your advice. In Figure 3, although the oil film pressure was measured by authors, the circumferential oil film pressure could be approximated as the measured oil film pressure.

  1. Response to comment: Is the variation of the axis whirl orbit of rotor in Figure 13 obtained from experiments or calculations? There is no time domain calculation method for the axis whirl orbit of rotor mentioned in the paper.

Response: Thank you for your suggestion. The variation of the axis whirl orbit of rotor in Figure 13 was obtained by calculations. In addition, this work focused on the stability of the axis whirl orbit of rotor system under different operating conditions, there is no need for time-domain calculation of the axial rotation trajectory of the rotor system. Therefore, the variation of eccentricity ratio with time has not been added by the author in this work.

We tried our best to improve the manuscript and made some changes in the manuscript. These changes will not influence the content and framework of the paper. And here we did not list the changes.

We appreciate for Editors/Reviewer’s warm work earnestly, and hope that the correction will meet with approval.

Once again, thank you very much for your comments and suggestions.

Reviewer 2 Report

Commented in the manuscript (pdf). Authors need to address all the points.

I do not see any novelty in the work as most of the conclusions are already known. The Governing equation (Reynolds Equation) does not incorporate the inertial effect due to consideration of turbulent flow. Velocity profile is also considered as parabolic, which is true for laminar flow consideration. In view of this, the presented results can not be relied upon.

The authors have considered Reynolds Boundary Condition (written wrongly), whereas mass conserving BC would have been appropriate. Authors should address this issue.

I do not recommend publication of the paper unless above fundamental issues are addressed.

English needs improvement.

Author Response

Dear Reviewer:

Thank you for your letter and for the your comments concerning our manuscript entitle” Analysis of the Turbulent Lubrication of a Textured Hydrodynamic Journal Bearing” (ID:lubricants-2532008). Those comments are all valuable and very helpful for revising and improving our paper, as well as the important guiding significance to our researches. We have studied comments carefully and have made correction which we hope meet with approval. The main corrections in the paper and the responds to the reviewer’s comments are as following:

Responds to the reviewer’s comments:

Reviewer #2:

  • Response to comment: The Governing equation (Reynolds Equation) does not incorporate the inertial effect due to consideration of turbulent flow?

Response: Thank you for pointing this problem. The Governing equation (Reynolds Equation)  does not incorporate the inertial effect has been revised by authors, the modified Reynolds equation involved the inertial effect has been supplemented in this paper, as detailed in expression 4.

  • Response to comment: What is Alpha?

Response: Thank you for pointing this problem. Alpha is the coefficient related to the bearing structural and working conditions, which should be obtained according to  experimental data. And, the detailed explanation has been supplemented by authors under the Equation 2 in Section 2.2.

  • Response to comment: P=0 and dP/dtheta=0 at theta=thetac thetac being on set of cavitation

Response: Thank you for pointing this problem. The content of oil film boundary condition has been revised by authors according to your suggestion, as detailed in section 2.3, line 116.

  • Response to comment: This velocity profile being parabolic pertains to Laminar regime. How turbulent regime is incorporated?

Response: Thank you for pointing this problem. I'm very sorry for causing your misunderstanding because authors failed to put the turbulence factors Gθ and Gλ into the expression 10, and the expression 10 has been revised by authors.

  • Response to comment: It is a well known fact that fluid inertia increases with higher load and speed as per definition. Then how does this observation require verification?

Response: Thank you for your valuable suggestion. Currently, a series of studies on the lubrication performance of textured hydrodynamic journal bearings under turbulent regime have been conducted based on turbulent model, and the contents and conclusions involved were based on turbulent model. Therefore, the observation that fluid inertia increases with the increase of load and speed does not need to be verified because this work is a theoretical study. Nevertheless, it is a good idea, which provides ideas for the author's further research work. However, the correctness and accuracy of the model have been verified by authors, as detailed in section 3.

  • Response to comment: I do not see any novelty in the work as most of the conclusions are already known?

Response: Thank you for your valuable suggestion. According to your suggestion, the conclusions without novelty has been deleted, and the conclusions has been summarized and rewritten in the part of conclusions by authors, as detailed in section 5.

We tried our best to improve the manuscript and made some changes in the manuscript. These changes will not influence the content and framework of the paper. And here we did not list the changes.

We appreciate for Editors/Reviewer’s warm work earnestly, and hope that the correction will meet with approval.

Once again, thank you very much for your comments and suggestions.

Reviewer 3 Report

This manuscript tries to investigate the lubrication performance of textured hydrodynamic journal bearing. The main innovation is the development of the turbulent model. Besides, both simulation and experiments results are shown and compared. The conclusions are clear. However, several detail is missing and needs to be explained more.

1.     The detail of the structure of the texture on the bearing need to be explained. What is the size of the dimple?The interval distance? The depth?

2.     For each figure, it is important to clear mention its origin: simulation or experiment. For example, For Fig.13,is it a simulation result or ax experimental results?

3.     In Fig.1, the model shows that the maximum pressure is around θ=300o, however, in Fig.3, the maximum pressure is around θ=135o. A unified coordinate system should be used.

4.     In Fig.6, the x axis is circumferential coordinate, what is the relation between the circumferential coordinate and the angular coordinate?

5.     In Fig.8(b), it was mentioned that is shows the oil film thickness distribution, however, the y axis is dimensionless pressure. Which one is right?

6.     In Fig.9(b-d), Fig.10(b) ,Fig11(b),part of the curves is obscured by the chart bar. In Fig(c-d), a smaller figure is at the top right corner. However, the curves are crossed. It is not clear using such a figure form.

Author Response

Dear Reviewer:

Thank you for your letter and for the your comments concerning our manuscript entitle” Analysis of the Turbulent Lubrication of a Textured Hydrodynamic Journal Bearing” (ID:lubricants-2532008). Those comments are all valuable and very helpful for revising and improving our paper, as well as the important guiding significance to our researches. We have studied comments carefully and have made correction which we hope meet with approval. The main corrections in the paper and the responds to the reviewer’s comments are as following:

Responds to the reviewer’s comments:

Reviewer #3:

  1. Response to comment: The detail of the structure of the texture on the bearing need to be explained. What is the size of the dimple?The interval distance? The depth?

Response: Thank you for pointing this problem. According to your suggestion, as mentioned in this paper, the physical meanings of some variables (such as, depth, interval distance and diameter et al. ) are shown in Table 1.

  1. Response to comment: For each figure, it is important to clear mention its origin: simulation or experiment. For example, For Fig.13, is it a simulation result or ax experimental results?

Response: Thank you for pointing this question. According to your suggestion, the variation of the axis whirl orbit of rotor in Figure 13 was obtained by calculations based on expression (13-15) and the its origin mentioned has been supplemented in section 4.4 by authors.

  1. Response to comment: In Fig.1, the model shows that the maximum pressure is around θ=300o, however, in Fig.3, the maximum pressure is around θ=135o. A unified coordinate system should be used.

Response: Thank you for the reviewer’s kind advice. According to your suggestion, Fig. 1 has been modified by the authors to use a unified coordinate system with Fig. 3, according to your suggestion.

  1. Response to comment: In Fig.6, the x axis is circumferential coordinate, what is the relation between the circumferential coordinate and the angular coordinate?

Response: Thank you for your advice. In fact, the angular coordinate represents angle, while circumferential coordinate represents arc length. There is a relationship between circumferential coordinate and angular coordinate, which is shown as angular coordinates =360/2*pi×circumferential arc length.

  1. Response to comment: In Fig.8(b), it was mentioned that is shows the oil film thickness distribution, however, the y axis is dimensionless pressure. Which one is right?

Response: Thank you for your suggestion. I'm very sorry for your misunderstanding due to authors negligence in writing. The expressions in this paper have been revised by the authors.

  1. Response to comment: In Fig.9(b-d), Fig.10(b), Fig11(b),part of the curves is obscured by the chart bar. In Fig9(c-d), a smaller figure is at the top right corner. However, the curves are crossed. It is not clear using such a figure form.

Response: Thank you for your valuable suggestion. According to your suggestion, the Fig.9 (b-d) in this work have been revised by authors.

We tried our best to improve the manuscript and made some changes in the manuscript. These changes will not influence the content and framework of the paper. And here we did not list the changes.

We appreciate for Editors/Reviewer’s warm work earnestly, and hope that the correction will meet with approval.

Once again, thank you very much for your comments and suggestions.

Round 2

Reviewer 1 Report

The authors have made detailed revisions to the writing of the paper, and  answered each comment. However, there are still some shortcomings in the author's description of the significance of conducting research on the lubrication performance of low-speed heavy water lubricated bearings in the Introduction section. The following articles may be helpful to the author:

(1)Study on nonlinear dynamic characteristics of propulsion shafting under friction contact of stern bearings.

(2)Tribological modification of hydrogenated nitrile rubber nanocomposites for water-lubricated bearing of ship stern shaft.

(3)Tribological modification of hydrogenated nitrile rubber nanocomposites for water-lubricated bearing of ship stern shaft.

Author Response

Dear Reviewer:

Thank you for your letter and for the your comments concerning our manuscript entitle” Analysis of the Turbulent Lubrication of a Textured Hydrodynamic Journal Bearing” (ID:lubricants-2532008). Those comments are all valuable and very helpful for revising and improving our paper, as well as the important guiding significance to our researches. We have studied comments carefully and have made correction which we hope meet with approval. The main corrections in the paper and the responds to the reviewer’s comments are as following:

Responds to the reviewer’s comments:

Reviewer #1:

Response to comment: The authors have made detailed revisions to the writing of the paper, and  answered each comment. However, there are still some shortcomings in the author's description of the significance of conducting research on the lubrication performance of low-speed heavy water lubricated bearings in the Introduction section.

Response: Thank you for your suggestion. Only two articles were found (on the study of nonlinear dynamic characteristics of propulsion shafting under frictional contact between stern bearing and second bearing), since the second was similarly with the third article. Therefore, the articles found have been added to this work by  authors, as detailed in reference [3] and [21]. 

We tried our best to improve the manuscript and made some changes in the manuscript. These changes will not influence the content and framework of the paper. And here we did not list the changes.

We appreciate for Editors/Reviewer’s warm work earnestly, and hope that the correction will meet with approval.

Once again, thank you very much for your comments and suggestions.

Reviewer 2 Report

Most of the issues raised by me are addressed by the authors. However, modified Reynolds equation should have been derived from original Navier-Stokes equation without neglecting inertial effect. Relevant published research, though limited, are available. The present analysis is, at the most, considered as a step forward only.

Author Response

Dear Reviewer:

Thank you for your letter and for the your comments concerning our manuscript entitle” Analysis of the Turbulent Lubrication of a Textured Hydrodynamic Journal Bearing” (ID:lubricants-2532008). Those comments are all valuable and very helpful for revising and improving our paper, as well as the important guiding significance to our researches. We have studied comments carefully and have made correction which we hope meet with approval. The main corrections in the paper and the responds to the reviewer’s comments are as following:

Responds to the reviewer’s comments:

Reviewer #2:

Response to comment: Most of the issues raised by me are addressed by the authors. However, modified Reynolds equation should have been derived from original Navier-Stokes equation without neglecting inertial effect. Relevant published research, though limited, are available. The present analysis is, at the most, considered as a step forward only.

Response: Thank you very much for your appreciation. In the following research, the authors will make a more in-depth study on the dynamic performance of the turbulent lubrication rotor-bearing system based on the modified Reynolds equation without neglecting inertial effect, according to your suggestion.

We tried our best to improve the manuscript and made some changes in the manuscript. These changes will not influence the content and framework of the paper. And here we did not list the changes.

We appreciate for Editors/Reviewer’s warm work earnestly, and hope that the correction will meet with approval.

Once again, thank you very much for your comments and suggestions.

Reviewer 3 Report

none

Author Response

Dear Reviewer:

Thank you for your letter and for the your comments concerning our manuscript entitle” Analysis of the Turbulent Lubrication of a Textured Hydrodynamic Journal Bearing” (ID:lubricants-2532008). Those comments are all valuable and very helpful for revising and improving our paper, as well as the important guiding significance to our researches. We have studied comments carefully and have made correction which we hope meet with approval. The main corrections in the paper and the responds to the reviewer’s comments are as following:

Responds to the reviewer’s comments:

Reviewer #3:

Response to comment: None.

Response: Thank you very much for your comments. 

We tried our best to improve the manuscript and made some changes in the manuscript. These changes will not influence the content and framework of the paper. And here we did not list the changes.

We appreciate for Editors/Reviewer’s warm work earnestly, and hope that the correction will meet with approval.

Once again, thank you very much for your comments and suggestions.